# Theoretical and Experimental Analysis of the Hot Torsion Process of the Hardly Deformable 5XXX Series Aluminium Alloy

**DOI:** 10.3390/ma14133508

**Published:** 2021-06-23

**Authors:** Konrad Błażej Laber, Beata Leszczyńska-Madej

**Affiliations:** 1Department of Metallurgy and Metal Technology, Faculty of Production Engineering and Materials Technology, Czestochowa University of Technology, 19 Armii Krajowej Ave., 42-201 Czestochowa, Poland; 2Faculty of Non-Ferrous Metals, AGH University of Science and Technology, 30-059 Kraków, Poland; bleszcz@agh.edu.pl

**Keywords:** hot torsion test, strain and stress state, numerical modelling, physical modelling, metallographic examinations, EBSD analysis, microhardness, inhomogeneity of microstructure, hardly deformable materials

## Abstract

This work presents the results of the numerical and physical modelling of the hot torsion of a hardly deformable 5XXX series aluminium alloy. Studies were conducted on constrained torsion with the use of the STD 812 torsion plastometer. The main purpose of the numerical tests was to determine the influence of the accuracy of the mathematical model describing the changes in the yield stress of the tested material on the distribution of strain parameters and on the stress intensity. According to the preliminary studies, in the case of numerical modelling of the torsion test, the accuracy of the applied mathematical model describing the changes in the rheological properties of the tested material and the correct definition of the initial and boundary conditions had a particularly significant impact on the correctness of the determination of the strain parameters and the intensity of stresses. As part of the experimental tests, physical modelling of the hot torsion test was conducted. The aim of this part of the work was to determine the influence of the applied strain parameters on the distribution and size of grain as well as the microhardness of the tested aluminium alloy. Metallographic analyses were performed using light microscopy and the electron backscatter diffraction method. Due to the large inhomogeneity of the deformation parameters and the stress intensity in the torsion test, such tests were necessary for the correct determination of the so-called representative area for metallographic analyses. These types of studies are particularly important in the case of the so-called complex deformation patterns. The paper also briefly presents the results of preliminary research and future directions in which it is planned to use complex deformation patterns for physical modelling of selected processes combining various materials.

## 1. Introduction

There are many research methods to determine the value of yield stress as a function of strain parameters and temperature. These methods include tensile, compression, and torsion tests. The methods have been described in detail in other works including [1,2,3,4,5]. According to many authors, the most convenient way of determining the flow curves in high temperature is a hot torsion test. It enables the determination of the yield stress indirectly, using the hypothesis of material effort. This test is widely used in plastometric tests due to the unchanging stress state during the tests, corresponding with high accuracy to pure shear and lack of friction as well as the possibility of obtaining large deformations, significantly exceeding deformations achievable in the tensile or compression test [1,2,6]. The hot torsion method is particularly advantageous when assessing the plasticity of hardly deformable and brittle materials for the conditions of hot plastic working. The advantage of this method is also the fact that during the tests, it is possible to create more secure conditions to obtain a constant strain rate and to model sequential (multistage) deformation patterns in the best possible way [1,7].

The main areas of application of the torsion test are determination of the value of yield stress at a given temperature and for a given stain value and strain rate, assessment of the limit deformability of the tested materials, testing (in conjunction with metallographic analyses) of structural changes in the material caused by the deformation process or physical modelling of plastic working processes [6].

There are many examples of the use of the hot torsion test in basic research in the literature [8,9,10,11,12,13,14]. In these works, the authors investigated, for example, the influence of the applied conditions of the deformation process on the yield stress, the development of the microstructure and its gradient in the cross-section, or the influence of the applied conditions on the deformability of the tested materials. While analysing the available literature, numerous papers can also be found in which hot torsion tests have been successfully used to physically copy the actual technological processes of rolling, extrusion or combining materials [15,16,17,18,19,20,21].

However, it should be noted that the hot torsion test is marked by an uneven strain distribution, strain rate, and temperature on the cross-section and longitudinal section of the torsion material. In the cross-section, all the parameters mentioned above reach their maximum values on the sample surface, while the lowest values of strain parameters occur in the material axis. The influence of this unevenness should be taken into account through calculating the strain parameters for the so-called representative radius, in which the values of the strain parameters correspond to the average values in the cross-section. The diversity of the strain value and the strain rate depending on the analytical relationships used, linking the equivalent strain with the non-dilatational strain, result in the assumption of different values, the so-called equivalent radius [2]. The projected equivalent radius values are: 0.6 r, 0.67 r, 0.724 r or 0.75 r, where r is the radius of the torsion sample [2,3,22].

Numerical modelling of the torsion test, conducted in accordance with the actual conditions of experimental tests, enables, for example, to determine the representative area for possible metallographic tests. On the basis of the obtained results, it is also possible to analyse the state of strain and stresses in the entire volume of the material. This is especially important for complex deformation patterns (e.g., simultaneous torsion with tension or simultaneous torsion with compression). As the preliminary tests conducted in this area show, the applied strain pattern affects the value of strain parameters (strain intensity and strain rate intensity) and the yield stress of the material. This is especially visible during simultaneous torsion and tension. It was initially determined that this is the result of the necking down formation and, consequently, a strong location (concentration) of the strain in this area. On the basis of the results of many experimental and theoretical studies, it has been proved that the strain rate in the neck during the tensile test is an order higher than the average strain rate calculated on the basis of changes in the measuring length of the samples [1].

Numerical analysis of the torsion test also enables the determination of the degree of unevenness of the strain parameters for different materials and different dimensions of the work part of the torsion samples in a wide range of strain parameters and temperature. However, the necessary condition for the correct determination of the strain and stress parameters during numerical modelling of the hot torsion test is the precise definition of the rheological properties of the tested material and the correct determination of the initial and boundary conditions, in accordance with the experimental tests.

An equally important problem during the theoretical and experimental analysis of the hot torsion test is the determination of the influence of the applied strain parameters, for example, on the distribution and size of the grain as well as the microhardness of the tested material. As shown in the literature, both the grain size and its distribution, as well as the microhardness distribution, depend primarily on the properties of the material itself, as well as on the parameters of the deformation process (caused strain value, strain rate and temperature) and the mechanisms (processes) of rebuilding microstructures [10,11,23,24,25]. The description of the most common mechanisms of remodelling the microstructure and the behaviour of various materials during hot forming can be found, for example, in the works [26,27,28,29,30,31]. This applies not only to the torsion processes but also to high-pressure torsion [32,33,34].

On the basis of the conducted numerical analyses, it was found that the condition for the correct determination of the strain parameters distribution and the stress intensity in the torsion test is high accuracy of the mathematical model describing the rheological properties of the material tested.

No studies have been found in the analysed literature that would comprehensively describe both the influence of the accuracy of the mathematical model of rheological properties on the values and nature of changes in the strain parameters and the yield stress as well as the influence of the applied strain parameters on the nature of the grain size distribution and the microhardness of 5xxx series aluminium alloy. Therefore, according to the authors, the issues raised in the work are topical, especially in the context of complex deformation patterns.

## 2. Materials and Methods

### 2.1. Materials and Process Parameters

The tests presented in the paper were conducted for the hardly deformable 5019 aluminium alloy, with the chemical composition compatible with the standard EN 573-3 presentedin details in the paper [35]. Plastometric tests were conducted up to the true strain value equal to 5, for the true strain rate value 0.05, 0.25 and 0.5 s^−1^, at the temperature 360, 400, 440 480 and 520 °C [35].

### 2.2. Methods

The methods for the numerical and physical modelling were used for the research. Numerical modelling was performed with the use of commercial FORGE 2011 software [36], while physical modelling was performed with the use of the STD 812 torsion plastometer [14]. Metallographic studies were conducted using light microscopy (Olympus GX 51 microscope, Tokyo, Japan) and electron backscatter diffraction (EBSD) (Hitachi S-3400N microscope, Tokyo, Japan). The microhardness measurements were carried out using the Innovatest Nexus 4000 microhardness tester (INNOVATEST Europe BV, Maastricht, The Netherlands).

#### 2.2.1. Numerical Research: Mathematical Model Using FORGE 2011^®^ Software

The aim of the numerical tests was to determine the influence of the accuracy of defining the rheological properties of the 5019 aluminium alloy on the strain intensity distribution, strain rate intensity, and stress intensity in the hot torsion test. In the first stage, plastometric tests of the analysed aluminium alloy were conducted in constrained torsion tests, the results of which are described closely in the paper [35]. These tests were carried out for conditions characteristic of the extrusion process. The obtained results were approximated with different accuracies (variants no. 1 and 2), determining mathematical models of the rheological properties of the tested material in the analysed range of strain and temperature parameters. In the next stage of the research, numerical modelling of the hot torsion test was conducted with the use of the developed models of changes in the rheological properties of the tested alloy, in accordance with the conditions occurring during experimental tests. These studies were carried out with the use of the FORGE 2011^®^ program [36].

In the FORGE 2011^®^ program [36], the properties of the deformed material were described according to the Norton–Hoff [37,38,39,40] conservation law written in the following Equation (1):(1)$$S_{ij}=2K(T,{\dot{\varepsilon }}_{i},\varepsilon_{i}){\left(\sqrt{3}{\dot{\varepsilon }}_{i}\right)}^{m_{m}-1}{\dot{\varepsilon }}_{ij}$$
where $S_{ij}$ is the deviatoric stress tensor, ${\dot{\varepsilon }}_{i}$ is the strain rate intensity ${\dot{\varepsilon }}_{ij}$ is the strain rate tensor, $\varepsilon_{i}$ is the strain intensity, T is the temperature, K is the consistency depending on the yield stress $\sigma_{p}$, and $m_{m}$ is the coefficient characterizing hot deformation of the metal (0 < $m_{m}$ < 1).

For the determination of the temperature field, a differential equation is used, which describes variations in temperature with transient heat flow. This is a quasi-harmonic equation in the following Equation (2) [41,42,43,44]:(2)$$\frac{\partial }{\partial x}\left(k_{x}\frac{\partial T_{s}}{\partial x}\right)+\frac{\partial }{\partial y}\left(k_{y}\frac{\partial T_{s}}{\partial y}\right)+\frac{\partial }{\partial z}\left(k_{z}\frac{\partial T_{s}}{\partial z}\right)+\left(Q-c_{p}\rho \frac{\partial T_{s}}{\partial t}\right)=0$$
where $k_{x}$,$k_{y}$ and $k_{z}$ are functions of the distribution of anisotropic thermal conductivities in the directions *x*, *y* and *z*, $T_{s}$ is the function that describes temperature in the investigated area,Q is the function of the distribution of deformation heat generation speed, $c_{p}$ is the function of the distribution of metal-specific heat and ρ is the function of the metal density distribution. 

When a die is rotating, the solver computes the torque obtained around the rotation axis. The rolling torque is equivalent to a force multiplied by a distance. So, the solver computes on every surface element the vector force multiplied with the distance to the axis of rotation. The sum of all these local torques provides the total torque as shown in Equation (3) [41,42]:(3)$$M_{w}=\int_{S}r\sigma_{y}dS=\sum_{e=1}^{N_{e}}\int_{S_{e}}r_{e}\sigma_{ye}l_{y}dS_{e}$$
where *r* is the distance between a point on the material surface and the rotation axis, *σ_y_* is the stress component, *S_e_* is the surface area of a finite element in contact with the tool, *e* is a finite element in contact with the tool, *N_e_* is the number of finite elements that remain in contact with the tool, *σ_ye_* is a component of the stress acting on the element in contact with the rotating tools and *l_y_* is the directional cosine of the normal to the contact area of the current element with the rotating tool.

The initial and boundary conditions for the numerical modelling of the hot torsion test were the same as for the experimental tests using the STD 812 torsion plastometer (Section 2.1 and Section 2.2.2). Friction conditions between the sample and the tools were set to bilateral sticking. To obtain a constant temperature distribution on the sample length, the thermal exchange was set as adiabatic.

In order to describe the changes in the value of the yield stress during the approximation of the results of the experimental tests, Equation (4) [43] was adopted:(4)$$\sigma_{p}=A\cdot e^{m_{1}\cdot T}\cdot T^{m_{9}}\cdot \varepsilon^{m_{2}}\cdot e^{\frac{m_{4}}{\varepsilon }}\cdot {\left(1+\varepsilon \right)}^{m_{5}\cdot T}\cdot e^{m_{7}\cdot \varepsilon }\cdot {\dot{\varepsilon }}^{m_{3}}\cdot {\dot{\varepsilon }}^{m_{8}\cdot T}$$
where $\sigma_{p}$ is yield stress, T is temperature, ε is true strain, $\dot{\varepsilon }$ is strain rate and A and m_1_÷m_9_ are function coefficients. 

The approximation of the results of the experimental tests was performed with the use of software developed at the Department of Metal Forming and Safety Engineering at the Częstochowa University of Technology.

#### 2.2.2. Experimental Research

As part of the experimental tests, physical modelling of the hot torsion test was conducted. The STD 812 torsion plastometer was used for the tests [14]. The general view of the test chamber and the main parameters of the device are shown in Figure 1.

Both the experimental and numerical tests were conducted to obtain the true strain value equal to 5, at a constant temperature of the deformed sample and with a constant strain rate. Round samples with a diameter of *d* = 8 mm and a measuring base length of *l* = 20 mm were used for the tests (Figure 2). A K-type thermocouple (NiCr–Ni), welded to the side surface of the sample, was used to register and control changes in temperature. The tested material was heated with a heating rate 5 °C/s to the desired temperature, held for 10 s, then deformed and cooled freely.

Equation (5) was used to determine the true strain, the true strain rate was determined based on Equation (6), while the yield stress was calculated according to Equation (7) [2,22].
(5)$$\varepsilon =\frac{2\cdot \pi \cdot r\cdot N}{\sqrt{3}\cdot L}$$
(6)$$\overset{•}{\varepsilon }=\frac{2\cdot \pi \cdot r\cdot \overset{•}{N}}{\sqrt{3}\cdot 60\cdot L}$$
(7)$$\sigma_{p}=\frac{\sqrt{3}\cdot 3\cdot M}{2\cdot \pi \cdot r^{3}}$$
where r is the sample radius, *L* is the sample length, *N* is the number of sample twists (revolutions), $\overset{•}{N}$ is the torsion speed (rpm), and *M* is the torque. 

#### 2.2.3. Metallographic Analysis and Microhardness Measurements

As part of the experimental work, metallographic tests and microhardness measurements of the tested 5019 aluminium alloy in the initial state were conducted after homogenisation before deformation and after the deformation process. The purpose of this part of the study was to determine the influence of the applied strain parameters on the distribution and size of the grain as well as the microhardness of the tested aluminium alloy for the selected temperature and strain rate. Observations of the microstructure were carried out using an Olympus GX51 light microscope equipped with digital image recording. Additionally, the grain size measurements on the cross-section were performed using the secant method.

Microhardness measurements were conducted using the Vickers method with the use of the Innovatest Nexus 4000 hardness tester, applying a 100 gramme load. Both the measurements of grain size and microhardness were carried out in two perpendicular directions on the cross-sections of samples made of the 5019 aluminium alloy (according to Figure 15). The test samples were mechanically polished according to the procedure of the Struers company. To reveal the microstructure, the samples were etched in Barker reagent with the composition of 2 mL HBF_4_ + 100 mL of H_2_O, and the observations were made under polarized light.

Metallographic analyses using the electron backscatter diffraction (EBSD) method were performed using a Hitachi S-3400N scanning electron microscope. An accelerating voltage of 15 kV was used for the tests. The sample was tilted at an angle of 70° at a distance of approximately 20 mm from the column. The detector and the software used for the EBSD analysis were from the HKL company. The electron backscatter diffraction analysis was performed to obtain maps of the crystal lattice orientation distribution on the surface of the test sample with a 1 µm step at a magnification of 200×. The colours obtained were assigned to the obtained orientations according to the basic IPF triangle. The results are presented both in the form of a map of the crystallographic orientation distribution on the sample surface and in the form of pole figures (PFs) along with inverse pole figures (IPFs). The texture results were also calculated for orientation intensity to both PF and IPF with 10 × 10° clustering. The disorientation map was also determined, considering the grain boundaries above 15° (marked in black) and the sub-grain boundaries above 5° (marked in blue). Measurements were made in three areas: (1) the centre of the sample, (2) at 0.67 r (2.68 mm from the centre), and (3) at 0.724 r (2.9 mm from the centre), where r is the sample’s radius.

## 3. Results

In order to determine the actual temperature distribution along the sample’s length, temperature measurements were performed using the contact method with the use of three K-type thermocouples (NiCr–NiAl) (Figure 3). Examples of the results obtained during the tests at the temperature of 480 °C are presented in Figure 4. It was stated that the highest temperature value (480 °C) appeared in the centre of the sample. At the end of the measurement base, the temperature was only slightly lower (below 3%) at 466 °C. On this basis, it was found that the heating and temperature control system of the STD 812 torsion plastometer ensured an even temperature distribution along the length of the measurement base of the torsion samples.

By analysing the temperature changes outside the working part of the samples, a much larger drop in temperature can be found. The main reason for the rapid decrease in temperature was the increasing cross-section of the samples and the change in heat exchange conditions. The test area of the samples was located in the centre of the induction coil heated to the set temperature, while the rest of the samples was outside the area of heat influence caused by the induction field. Additionally, heat was conducted towards the cold grips of the torsion plastometer.

### 3.1. Analysis of the Numerical Research Results: Problem with the Mathematical Model’s Accuracy

Table 1 presents the values of the equation coefficients (4), approximating the test results of the 5019 aluminium alloy obtained in the tested range of the strain and temperature parameters.

Sample diagrams of the course of yield stress during the torsion of samples at the temperatures of 440 °C, 480 °C, and 520 °C are presented in Figure 5.

By analysing the course of the actual and approximate hardening curves of the analysed aluminium alloy, it can be stated that the values of yield stress, approximated by Equation (4) using the coefficients in Table 1, were close to the values of yield stress determined experimentally.

Despite the small range of the strain rate (0.05–0.5), it can be noticed that during the torsion of the tested alloy with a strain rate of 0.05 s^−1^, at the temperatures of 440 and 480 °C, the values of yield stress obtained as a result of the approximation, above the actual strain value of about three, are lower than the values determined experimentally. During the torsion of the tested alloy with a strain rate of 0.05 s^−1^, at the temperature of 440 °C, the maximum differences between the values of yield stress determined experimentally and obtained as a result of the approximation occurred at the true strain of five and amounted to over 23%. When deforming samples with a strain rate of 0.05 s^−1^, at a temperature of 480 °C, the maximum differences between the values of yield stress determined in torsion tests and the approximated values were about 21% (also for true strain of five). Similar differences between the actual and approximate values of yield stress were found on the basis of the analysis of the test results obtained during the torsion of the samples with a strain rate of 0.25 s^−1^ at temperatures of 480 and 520 °C. Moreover, in this case, the greatest differences between the actual and approximate values of the yield stress were noted for the true strain equal to five. During the deformation of the tested 5019 aluminium alloy at a temperature of 480 °C, they were approximately 22%, while during the torsion at a temperature of 520 °C, they were approximately 19%.

After implementing the determined coefficients of the mathematical model of changes in the rheological properties of the 5019 aluminium alloy to the material base of the FORGE 2011^®^ program, numerical modelling of torsion tests (Variant 1) was performed in the next stage of the research. Examples of the results obtained during the torsion of samples with a strain rate of 0.25 s^−1^ at a temperature of 480 °C are presented in Figure 6, Figure 7, Figure 8 and Figure 9. The analysis covers the temperature distribution, strain intensity, strain rate intensity and stress intensity.

Figure 6 shows the temperature distribution in the tested aluminium alloy, determined numerically.

By analysing the data presented in Figure 6, it was found that the initial temperature distribution in the material set at the beginning of the torsion test (Figure 6a) does not change during the entire hot torsion process (Figure 6b). The obtained distribution is consistent with the results obtained experimentally (Figure 4). By analysing the temperature distribution in the central part of the sample, on the cross-section (Figure 6c), it was found that the highest temperature values caused by the highest strain value occurred on the surface. In the tested case, the temperature difference on the cross-section of the tested alloy was 5 °C. Based on the analysis of the results of the temperature distribution, it can be stated that the conditions for conducting the tests at a constant temperature were met.

The distribution of strain intensity, strain rate intensity, and stress intensity in the entire volume of the working part of the tested material is presented in Figure 7, Figure 8 and Figure 9.

On the basis of the data presented in Figure 7, Figure 8 and Figure 9, inhomogeneity of the analysed parameters can be observed both on the cross-section and longitudinal sections of the deformed samples. The numerically determined values of the strain intensity and strain rate intensity were higher than the values obtained in the experimental tests for all the equivalent analysed radii. In the analysed case, the numerically calculated values of the strain intensity ranged from 6 (for the equivalent radius of the 0.6 r sample) to 7.2 (for the radius of 0.75 r), while in the experimental tests, the true strain was equal to five. The values of the strain rate intensity ranged from 0.6 s^−1^ (for the equivalent radius of the 0.6 r sample) to 0.78 s^−1^ (for the radius of 0.75 r), while in the experimental tests, the strain rate was 0.25 s^−1^. By analysing the results of the stress intensity distribution, it was found that the nature of its distribution was incorrect. The highest values of this parameter occurred for the equivalent sample radius of 0.6 r and not on the surface, which is inconsistent with the data published in the literature. On the basis of the obtained test results, it was found that the direct cause of incorrectly calculated strain intensity distributions, strain rate intensity and stress intensity as well as higher values of these parameters compared to the experimental tests may be an inaccurate description of the rheological properties of the tested alloy with the use of the coefficients presented in Table 1. For the analysed case (strain rate of 0.25 s^−1^, temperature of 480 °C), the approximated values of the yield stress used in the numerical tests were lower than the values obtained in the experimental tests. Consequently, the plastic deformation resistance was also lower. As a result, the numerically determined values of strain intensity and strain rate intensity were overestimated in relation to the values obtained in actual torsion tests.

Therefore, in the next stage of the work, the results of plastometric tests were re-approximated using the equation coefficients (4), obtained by approximation for a narrow temperature range (Table 2, Variant 2). The obtained results are presented graphically in Figure 10.

Based on the analysis of the data presented in Figure 10, it can be found that by increasing the accuracy of the approximation (coefficients m3, m4, m5 and m7 of the approximating Equation (4)), the accuracy of the mathematical description (model) of the rheological properties of the tested aluminium alloy was increased. In the diagrams, no differences were observed between the approximated and true values of yield stress for a strain rate of 0.05 s^−1^ during the torsion at the temperatures of 440 and 480 °C and for a strain rate of 0.25 s^−1^ during the torsion at the temperatures of 480 and 520 °C, which were observed using the coefficients of the mathematical model of rheological properties presented in Table 1.

In the next stage of the research, numerical modelling of the torsion tests was conducted again using the coefficients of the mathematical model of the rheological properties presented in Table 2. Similar to the previous case, the results obtained during the torsion of the samples with a strain rate of 0.25 s^−1^ at the temperature of 480 °C are shown in Figure 11, Figure 12 and Figure 13. As the results of the temperature distribution were similar to those presented in Figure 6, only the results concerning the strain intensity distribution, strain rate intensity and stress intensity in the entire volume of the working part of the tested alloy are presented below.

Based on the analysis of the data presented in Figure 11, it can be found that during the torsion of samples from the tested aluminium alloy with a strain rate of 0.25 s^−1^ at the temperature of 480 °C and up to the true strain value of five, the numerically determined values of the strain intensity changed from 4.3 (for the equivalent sample radius of 0.6 r) up to 5.1 (for the radius of 0.75 r). The highest accuracy between the values determined numerically and those obtained in experimental tests was obtained for the equivalent radii of the 0.67 r and 0.724 r samples.

Figure 12 presents that for the analysed case (strain rate of 0.25 s^−1^, temperature of 480 °C, true strain of five), numerically determined values of strain rate intensity ranged from 0.25 s^−1^ (for the equivalent sample radius of 0.6 r) to 0.3 s^−1^ (for the radius of 0.75 r). The highest accuracy between the numerically determined values and those obtained in the experimental tests was obtained for the equivalent sample radius of 0.6 r. For the remaining equivalent radii, the numerically determined value of the strain rate intensity was slightly higher than the value obtained in the experimental tests. For the analysed case, during the experimental tests, the speed of the moving tool (grip), after reaching the value of 20.5 rpm, was slightly fluctuating, while during the numerical tests it was at a constant level. This could cause a slight overestimation of the numerically determined intensity of the strain rate.

By analysing the data on the stress intensity distribution (Figure 13), it can be found that during the torsion of the samples from the tested material with a strain rate of 0.25 s^−1^, at the temperature of 480 °C, up to the true strain value equal to five, the numerically determined values of the stress intensity ranged from 44 MPa (for the equivalent sample radius of 0.6 r) to 47 (for the radius of 0.75 r). The highest accuracy between the numerically determined values and those obtained in the experimental tests was obtained for the equivalent sample radius of 0.6 r. For the remaining equivalent radii, the numerically determined stress intensity values only slightly differed from the yield stress obtained during the torsion tests. The reason for this, as before, may be the way of defining the rotational speed of the movable tool (grip) during numerical calculations. The numerically higher value of the strain rate intensity resulted in a slight increase in the stress intensity of the tested alloy.

By analysing the obtained results, it can be found that after applying the coefficients of the mathematical model of the rheological properties listed in Table 2, the numerically determined values of the analysed parameters corresponded with high accuracy to the values obtained in the true torsion tests. This confirms the significant influence of the accuracy of the mathematical model of the rheological properties on the distribution of the analysed parameters.

The correctness of the conclusions of the authors is also confirmed by the results of the experimentally and numerically calculated torsion moment as well as the results of the course of the yield stress (experimental and calculated on the basis of the numerically determined torsion moment values, using the dependency (7)), which are presented in Figure 14.

By analysing the data presented in Figure 14, it is possible to observe a characteristic peak in the value of the torsion moment and yield stress at the beginning of the torsion process, and then their decrease along with the increase in the deformation. Based on the analysis of the results of the torsion moment and yield stress, it was found that greater consistency between the values calculated and recorded during torsion tests was obtained by using the coefficients of the rheological properties model presented in Table 2 (Variant 2).

Based on the analysis of the results of the numerical tests, it was found that as representative radii, marking the area in which the values of strain and stress parameters corresponded to the average values on the cross-section of the torsion samples and, at the same time, to the values obtained in the experimental tests, the radii equal to 0.67 r and 0.724 r should be taken.

### 3.2. Analysis of the Experimental Research Results

Figure 15 shows the general diagram of the sample with the directions marked on the cross-section in which metallographic tests and microhardness measurements were carried out.

Figure 16 presents photos showing the microstructure of the tested alloy in its initial state, before the deformation process. Figure 17 presents sample photos of the microstructure showing the centre (Figure 17a) and the edge (Figure 17b) of the sample.

Figure 18 presents a graph showing the changes in grain size on the cross-section of the tested samples in their initial state.

On the basis of the metallographic analysis, it was found that the tested alloy in its initial state (after the homogenisation process) had a homogeneous microstructure, both on the cross-section and longitudinal section. The revealed grains had a regular shape. The determined average grain diameter for the initial material before the deformation process was approximately 68 µm (Figure 18).

Figure 19 and Figure 20 present sample photos showing the microstructure of the tested aluminium alloy after plastic deformation at the temperature of 480 °C with a strain rate of 0.25 s^−1^.

Figure 21 presents a graph showing the changes in the grain size on the cross-section of the tested 5019 aluminium alloy samples after the deformation process.

On the basis of the metallographic analysis, a large diversity of the microstructure on the cross-section and longitudinal section of the tested material was found. While analysing the results of the measurements of the grain size of the tested aluminium alloy on the cross-section after the hot torsion process at the temperature of 480 °C with a strain rate of 0.25 s^−1^, its typical distribution for the torsion test was found. The largest grains were located in the central area (sample axis), where the strain intensity value was the smallest (3.20). The size of single grains in this area reached even 120–140 µm, while the average grain size was approximately 47 µm. For the radius of 0.67, the average grain size was about 27 µm, and for the radius of 0.724, the average grain size was approximately 26 µm. In the surface area, where the value of the strain intensity was the highest (6.65), small recrystallised grains with an average size of approximately 18 µm were visible. Additionally, in this area, coagulated separations arranged in stripes were observed. It was found that as a result of the deformation, the average grain size of the tested alloy decreased by approximately 21 µm (31%) in the sample axis, 41 µm (60%) for the 0.67 r radius and 42 µm (62%) for the 0.724 r radius. The average grain size of the 5019 aluminium alloy decreased by 50 µm (74%).

In turn, the determined average grain diameter of the tested alloy after the cross-sectional deformation was approx. approximately 32 µm.

On the basis of the conducted metallographic analyses, it was shown that the structure was highly diversified on the cross-section of the sample of the tested material after the hot torsion, mainly due to the high inhomogeneity of the deformation parameters (strain intensity, strain rate intensity). The obtained results of metallographic analyses confirm the necessity to define the so-called representative area for the assessment of the microstructure of the material after the torsion process. The presented results also indicate the need to analyse the microstructure of twisted samples in a strictly defined zone for which the value of local strain should be determined.

In the next stage of the research, microhardness measurements were performed on the cross-section of the samples. Exemplary results of the microhardness measurement on the cross-section of the samples of the tested aluminium alloy before and after the deformation process are presented in Figure 22.

The obtained results did not show any differences in the level of microhardness for the material after homogenisation and after the hot torsion process, despite significant differences in the grain size on the cross-section. The average measured microhardness was approximately 76 HV0.1 (Figure 22).

However, diversity in microhardness on the cross-section of the tested samples were noted. The material after the plastic deformation process was marked by greater inhomogeneity of the microhardness distribution on the cross-section, which was mainly caused by a greater differentiation of the grain size on the cross-section. The nature of the microhardness distribution is consistent with the results of the grain size measurement. In the deformed axis of the sample, where the grain is the largest, the tested material had the lowest microhardness. On the other hand, in the surface areas, where the average grain size was the smallest, the average values of the microhardness of the tested alloy were the highest.

In the conditions of hot deformation of aluminium alloys, the strengthening effect resulting from deformation is counteracted by softening as a result of structural renewal processes. Therefore, despite the significant fragmentation of the microstructure, especially outside the centre, no increase in the level of microhardness was found.

During thermomechanical processing, mechanical properties of materials are affected by strain history, chemical composition and the microstructure. When aluminium alloys are strained at the elevated temperatures, they may experience work hardening and flow softening which result from dynamic recovery (DRV), dynamic recrystallisation (DRX) or dynamic precipitation transformations [47,48,49]. Due to the high stacking fault energy of aluminium alloys, dynamic recovery (DRV) was more likely to occur than dynamic recrystallisation (DRX) [50].

In order to characterize the microstructure of the samples more fully, studies were conducted using the EBSD. Since the radii assumed that the representative radii were equal to 0.67 r and 0.724 r, EBSD analyses were performed in these areas and additionally along the axis of the samples.

The EBSD analyses for the sample after homogenisation (starting material before the deformation process), as presented in Figure 23, were similar for each of the analysed areas. Basic triangles indicate the dominance of directions around the <101> pole, and the maximum intensity of the texture was 2.32–2.84. Pictures presenting the types of edge indicate the dominance of large angles, above 15°, while in the axis of the sample, large angles of disorientation constitute 85% of the studied population, and at a distance of 0.67 r and 0.724 r—over 90%. The size of grains/sub-grains in the area covered by the study determined on the basis of EBSD tests was 57 µm, and the average surface area was 3528 µm^2^. The determined grain size at a distance of 0.67 r and 0.724 r was slightly smaller, and it was approximately 55 µm for both distances.

The results of the EBSD analysis of the samples after hot torsion confirmed the earlier observations about the significant diversity of the microstructure on the cross-section (Figure 24 and Figure 25).

In the case of analyses performed on the axis of the sample, the directions around the <001> pole were found to be dominant. The maximum texture intensity was 4.18. Almost 40% of the measured grain edge population were small disorientation angles below 15°.

In the distance from the sample axis 0.67 and 0.724 r, fine, almost equiaxial grains were observed. Compared to the central part of the sample, the proportion of the large angle edges was above 15°; it was much larger and represented more than 75% of the measured grain edge population. At a distance of 2.68 mm (0.67 r) from the centre of the sample, the texture intensity distribution was centred around the <111> pole, the maximum intensity was 1.65. In turn, at a distance of 2.9 mm (0.724 r) from the centre, the intensity distributions were concentrated around the <112> and <111> poles, and the maximum intensity was 1.25.

The average size of grains and sub-grains measured along the axis of the sample using the EBSD method was approximately 20 µm (cross-sectional area, respectively: 840 µm^2^). At a distance of 0.67 and 0.724 r from the axis of the sample, the size of grains/sub-grains was comparable and amounted to approximately 9 µm, and the cross-sectional area was approximately 90 µm^2^.

The differences in the obtained results of the grain size measurements with the use of photos taken using light microscopy techniques and determined using the EBSD method were largely due to the fact that in the set of the results obtained with the EBSD technique, there were also sub-grains with limits below 15°. Low-angle edges were not distinguished while observing using the light microscopy technique.

## 4. Directions of Future Studies

Knowledge of the exact distribution of strain and stress parameters in the material is of particular importance in the case of the so-called complex deformation diagrams that are planned in the future for the physical modelling of selected combining processes of various materials. Figure 26 presents the preliminary results of the research on the applicability of complex deformation states for physical modelling of combining materials.

The test parameters were similar to those occurring during the friction welding process. The combining process was conducted in two stages at a temperature of 400 °C. In the first stage of the research, the material was twisted with compression for 3 s.

Sample diagrams of changes in parameters during the first stage of the friction welding process are presented in Figure 27, Figure 28 and Figure 29.

As can be seen from the data presented in Figure 27, Figure 28 and Figure 29, during this stage of the tests, the temperature of the tested material increased by approximately 40 °C, caused mainly by friction between the surfaces of the combined parts of the samples. The displacement along the sample axis was 2 mm, while the torsion angle was 8640° (24 turns). The strain caused by torsion was, in this case, approximately 45, while the strain rate was approximately 15 s^−1^. By analysing the course of changes in stress caused by simultaneous torsion and compression, its rapid increase in the initial phase of this stage can be observed. In the further part of the simultaneous compression and torsion, the stress increase was small or remains constant. The reason for this may be additional slip mechanisms (shear lines) caused by a complex load pattern (surely, it requires additional metallographic tests). As shown in the research, the complex load condition also reduced the pressure force at this stage of the friction welding process.

In the second stage of the combining process, the tested aluminium alloy was subjected to compression (3 mm) as shown in Figure 30 and Figure 31.

At this stage, a slight decrease in the temperature of the tested material was also observed. The reason for lowering the temperature at this stage of the friction welding process may be a low value of the true strain and the lack of friction. On the other hand, an increase in both the stress and the pressure force was observed. The increase in the pressure force at this stage results from the fact that, in this stage, the sample was only compressed; there were no additional slip lines lowering the energy and strength parameters.

The parameters used enabled a permanent combination of the material. Therefore, it was found that it is possible to physically model the process of combining materials using complex deformation patterns using the STD 812 plastometer. However, in order to develop an accurate methodology for combining materials and physical modelling of actual technological processes (e.g., friction welding, rolling of bimetallic materials), numerical and physical modelling tests should be conducted in a wide range of strain parameters. It is necessary to study, for example, the distribution of strain components depending on the applied load condition (strain state) and the influence of the parameters used on the plastic flow of the material and, thus, on the microstructure and mechanical properties (deriving from the quality of the combination). Then, the possibility of adapting the obtained results to actual technological processes should be checked. It may also be necessary to plan a different geometry of the samples.

According to the knowledge of the author, this type of research has not been conducted in Poland so far. Developing the theoretical and experimental methodology of joining various materials with the use of complex load states will enable an accurate prediction of process parameters that guarantee a permanent combination of individual components, which will significantly reduce costs and accelerate the implementation of the technology of combining new materials in industrial conditions.

## 5. Discussion and Conclusions

The obtained test results showed a significant impact of the accuracy of the mathematical model of rheological properties on the distribution and values of strain parameters and on the stress intensity of the tested aluminium alloy. The correct determination of the strain parameters and the stress intensity in the material is particularly important during the numerical analysis of complex strain states and, in the case of tests, the parameters of which exceed the scope of the research possibilities of the used equipment.

After conducting the research and after analysing the obtained results, the following conclusions can be drawn:The condition for the correct determination of the strain parameters’ distribution and the stress intensity in the torsion test is the high accuracy of the mathematical model describing the rheological properties of the tested material and the correct determination of the initial and boundary conditions, consistent with the experimental tests;As representative radii, defining the area in which the values of strain and stress parameters correspond to the average values on the cross-section of the torsion samples and, at the same time, to the values obtained in experimental tests, radii equal to 0.67 r and 0.724 r can be taken;Using the coefficients of the rheological properties model presented in Table 2, a high agreement was obtained between the recorded and the calculated values of the torsion moment and the yield stress;The heating and temperature control system installed in the STD 812 torsion plastometer ensures an even temperature distribution along the length and cross-section of the measurement base of the torsion samples;As a result of deformation in the hot torsion process, a significant fragmentation of the microstructure of the tested alloy was obtained. It was proved that the grain size distribution on the sample cross-section was inhomogeneous and typical for a torsion test. The diversity of the microstructure on the cross-section after the hot torsion results from the inhomogeneity of the strain parameters such as strain intensity, strain rate intensity and stress inhomogeneity. The largest grains occurred in the sample axis, where the strain intensity value was the smallest, and the smallest in the surface area, where the strain intensity value reached its maximum value;No significant differences were found in the level of the microhardness of the material after homogenisation and after the hot torsion process. However, it has been shown that the distribution of microhardness on the sample cross-section after the hot torsion was more inhomogeneous, which is caused by a greater differentiation of the grain size on the cross-section. The lack of diversity in the level of microhardness is related to the fact that under the conditions of hot deformation of aluminium alloys, the strengthening effect caused by plastic deformation was eliminated by softening as a consequence of the structural renewal processes;The results of the EBSD analysis after the hot torsion present that the sample axis was dominated by the directions around the <001> pole. At a distance of 2.68 mm from the centre of the sample (0.67 r), the distribution of the texture intensity was focused around the <111> pole, while at a distance of 2.9 mm from the centre (0.724 r), the intensity distributions were centred around the <112> and <111> poles. In addition, it was proved that the fraction of large disorientation angles increased along with the distance from the sample axis, and the share of large disorientation angles above 15° was comparable for the distance from the sample axis of 2.68 mm (0.67 r) and 2.9 mm (0.724 r) and amounted to about 75% in the axis of the sample, large angles of disorientation constituted 60% of the studied population;The obtained results of metallographic analyses confirm the necessity to define a representative area for the assessment of the microstructure of the material after the torsion process as well as indicate the need to analyse the microstructure of the torsion samples in a strictly defined zone, for which local values of the strain intensity, strain rate intensity and stress intensity should be determined.

## Figures and Tables

**Figure 1 materials-14-03508-f001:**
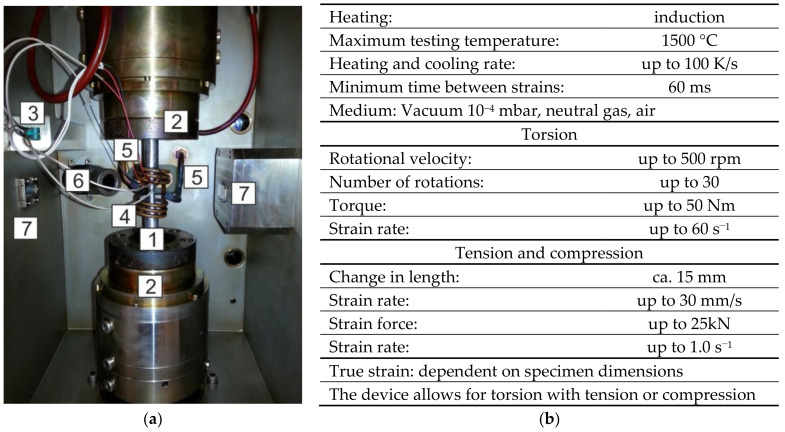
STD 812 torsion plastometer: (**a**) device chamber: 1—specimen, 2—holders, 3—thermocouples type S, 4—induction solenoid, 5—cooling system jets, 6—pyrometer and 7—sensors for laser measurement of specimen diameter; (**b**) basic specification.

**Figure 2 materials-14-03508-f002:**
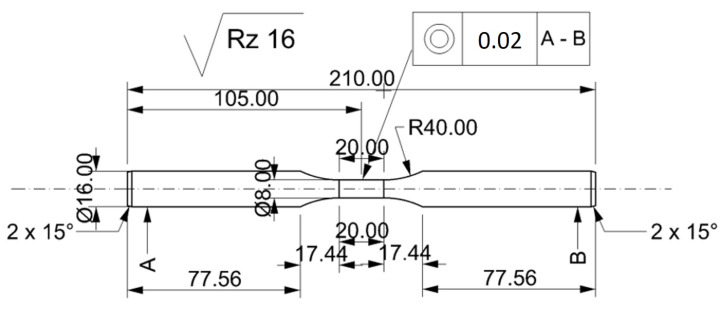
The 5019 aluminium alloy sample’s technical specification.

**Figure 3 materials-14-03508-f003:**
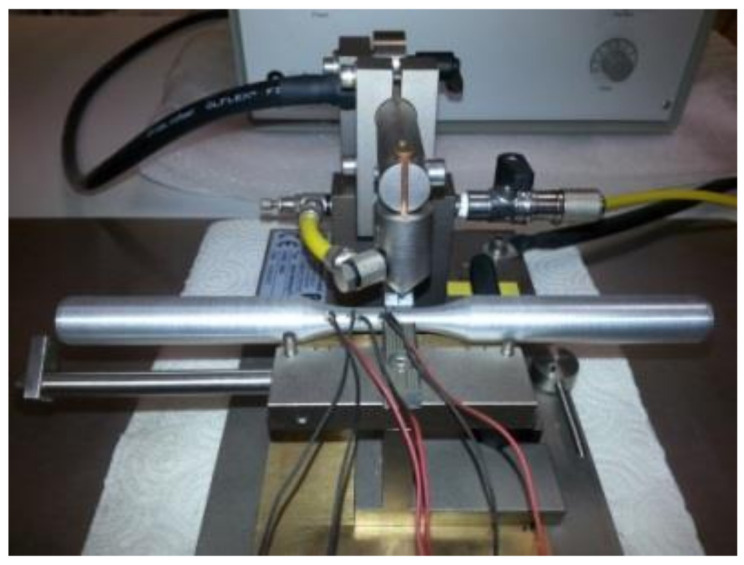
Example specimen during welding of the thermocouple with a lateral surface.

**Figure 4 materials-14-03508-f004:**
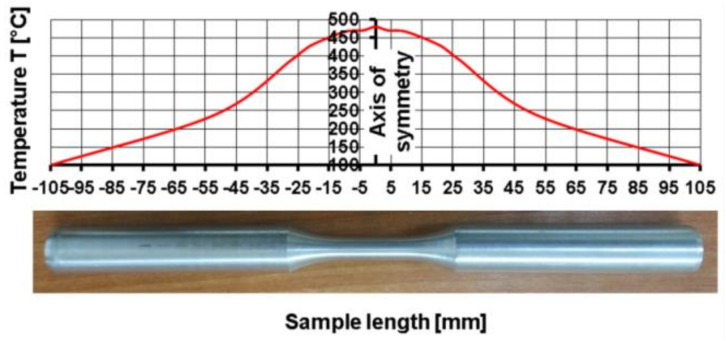
Actual temperature distribution along the sample’s length determined by the contact method.

**Figure 5 materials-14-03508-f005:**
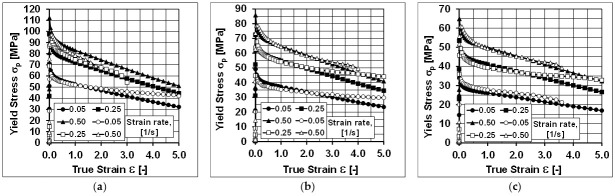
Flow curves of the 5019 aluminium alloy at temperatures of (**a**) 440 °C; (**b**) 480 °C; (**c**) 520 °C; empty symbols—plastometric test data; full symbols—results after approximation using coefficients from Table 1 (Variant 1).

**Figure 6 materials-14-03508-f006:**
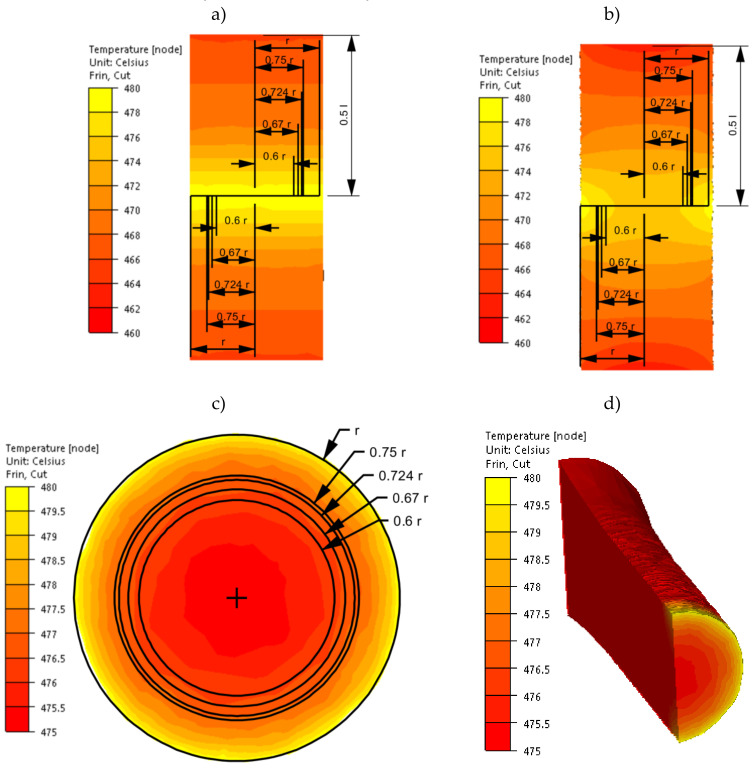
Temperature distribution of the 5019 aluminium alloy calculated numerically using coefficients from Table 1: (**a**) at the beginning of the torsion—longitudinal section, (**b**) at the end of the torsion—longitudinal section, (**c**) at the end of the torsion—cross-section in the centre of the working part of the sample and (**d**) at the end of the torsion—perspective.

**Figure 7 materials-14-03508-f007:**
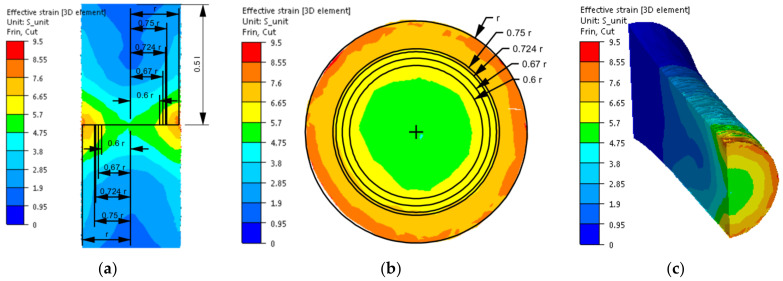
Distribution of the strain intensity of the 5019 aluminium alloy calculated numerically using coefficients from Table 1: (**a**) longitudinal section, (**b**) cross-section—centre of the working part and (**c**) perspective.

**Figure 8 materials-14-03508-f008:**
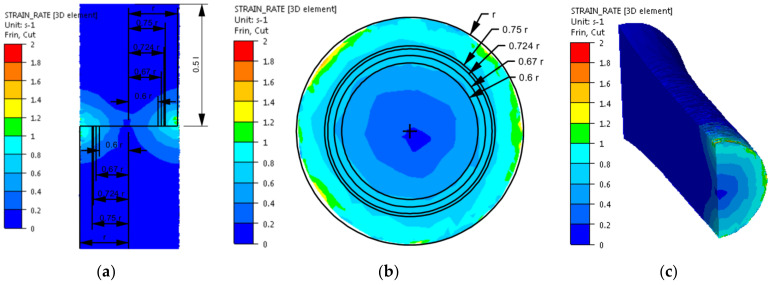
Distribution of the strain rate intensity of the 5019 aluminium alloy calculated numerically using coefficients from Table 1: (**a**) longitudinal section, (**b**) cross-section—centre of the working part and (**c**) perspective.

**Figure 9 materials-14-03508-f009:**
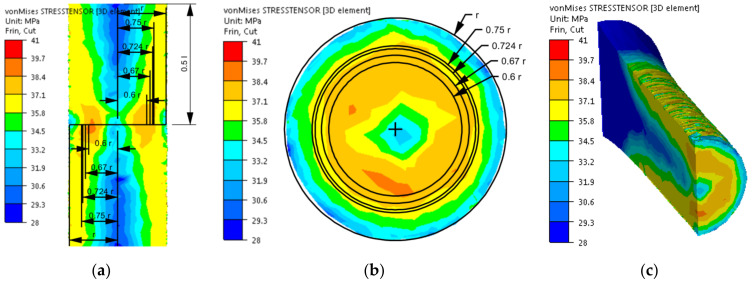
Distribution of the stress intensity of the 5019 aluminium alloy calculated numerically using coefficients from Table 1: (**a**) longitudinal section, (**b**) cross-section—centre of the working part and (**c**) perspective.

**Figure 10 materials-14-03508-f010:**
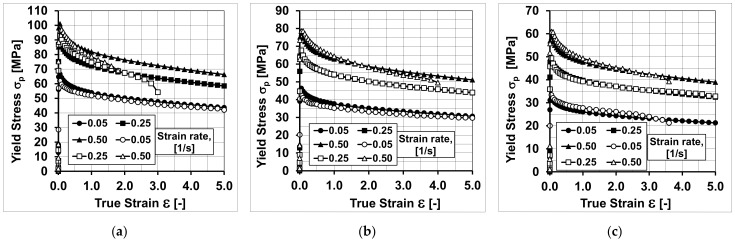
Flow curves of the 5019 aluminium alloy at temperatures of (**a**) 440 °C; (**b**) 480 °C; (**c**) 520 °C; empty symbols—plastometric test data; full symbols—results after approximation using coefficients from Table 2 (Variant 2).

**Figure 11 materials-14-03508-f011:**
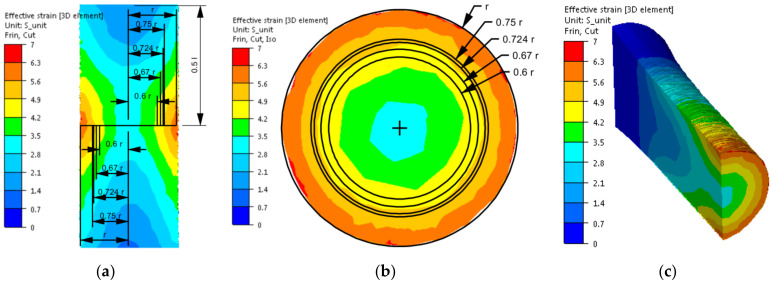
Distribution of the strain intensity of the 5019 aluminium alloy calculated numerically using coefficients from Table 2: (**a**) longitudinal section, (**b**) cross-section—centre of the working part and (**c**) perspective.

**Figure 12 materials-14-03508-f012:**
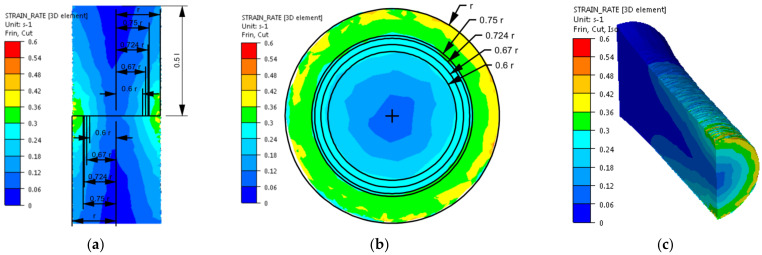
Distribution of the strain rate intensity of the 5019 aluminium alloy calculated numerically using coefficients from Table 2: (**a**) longitudinal section, (**b**) cross-section—centre of the working part and (**c**) perspective.

**Figure 13 materials-14-03508-f013:**
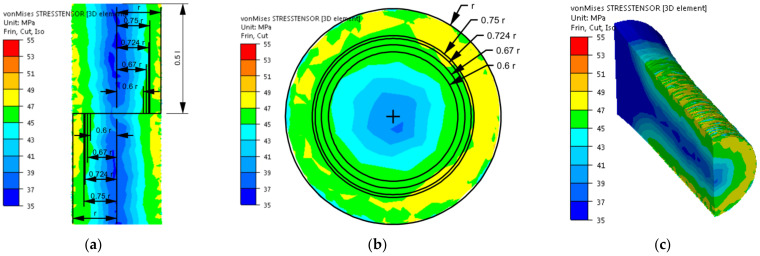
Distribution of the stress intensity of the 5019 aluminium alloy calculated numerically using coefficients from Table 2: (**a**) longitudinal section, (**b**) cross-section—centre of the working part and (**c**) perspective.

**Figure 14 materials-14-03508-f014:**
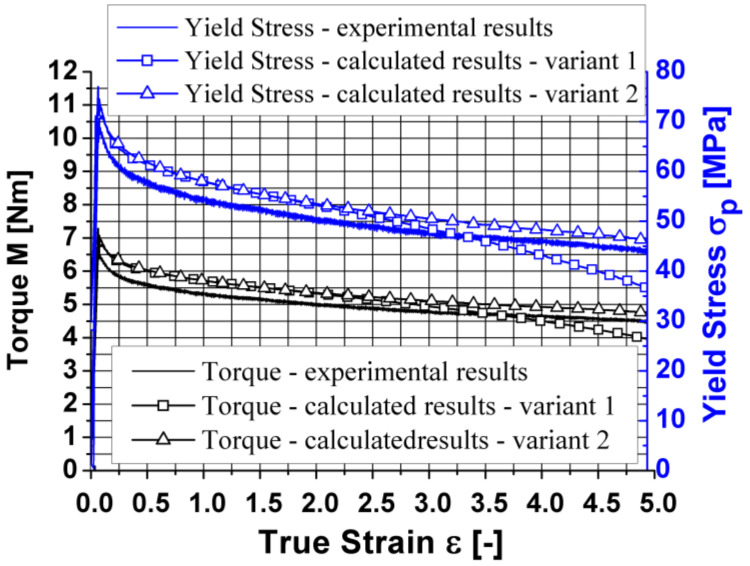
Changes in the torque and yield stress of the 5019 aluminium alloy—true and calculated values.

**Figure 15 materials-14-03508-f015:**
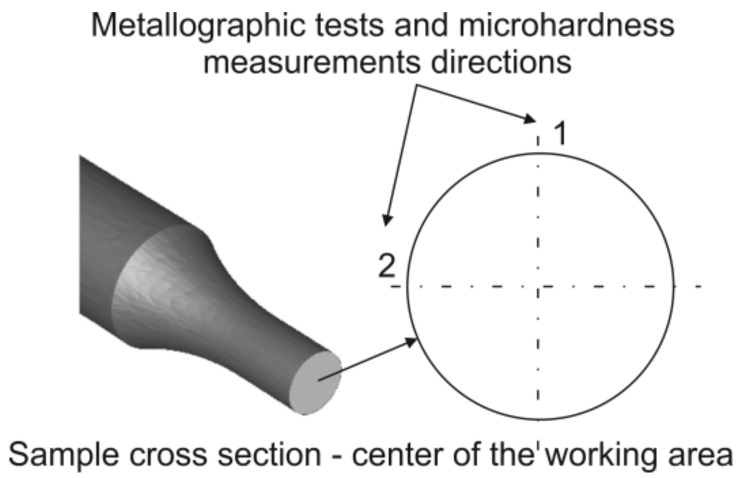
Cross-section of the sample with marked directions in which metallographic tests and microhardness measurements were carried out (general diagram).

**Figure 16 materials-14-03508-f016:**
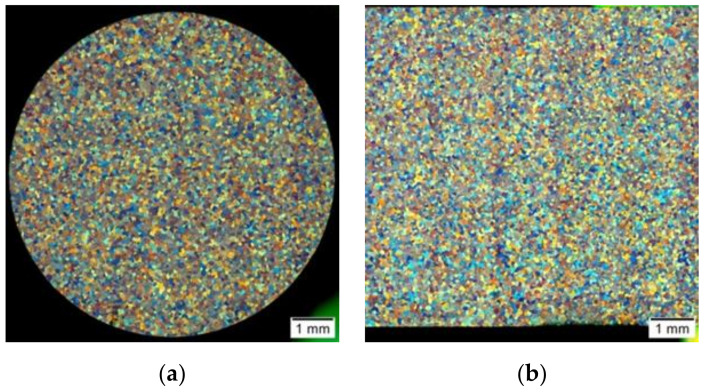
Microstructure of the 5019 aluminium alloy in initial state, after the homogenisation process (before the deformation process): (**a**) cross-section; (**b**) longitudinal section.

**Figure 17 materials-14-03508-f017:**
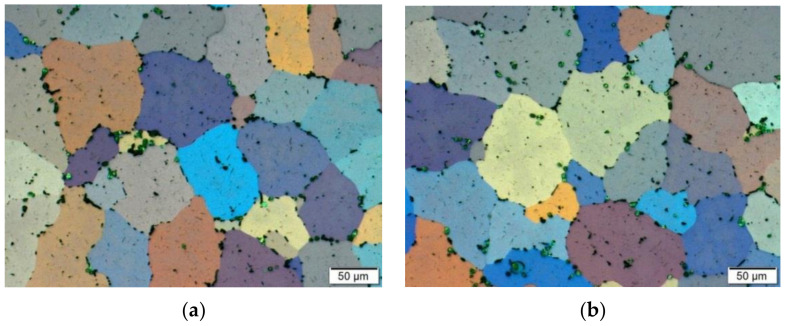
Microstructure of the 5019 aluminium alloy in the initial state, after the homogenisation process (before the deformation process): (**a**) sample centre; (**b**) sample edge; cross-section, magnification 200×.

**Figure 18 materials-14-03508-f018:**
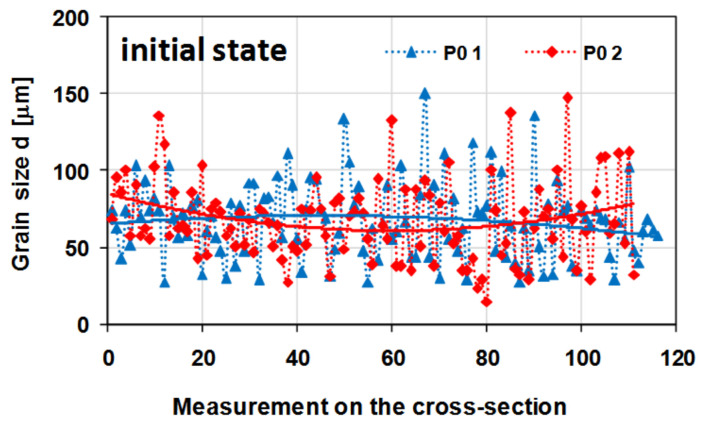
Changes in the grain size on the cross-section of the sample made of the 5019 aluminium alloy; the measurements were performed in two perpendicular directions (as shown in Figure 15); P0—the initial state of the material (after homogenisation).

**Figure 19 materials-14-03508-f019:**
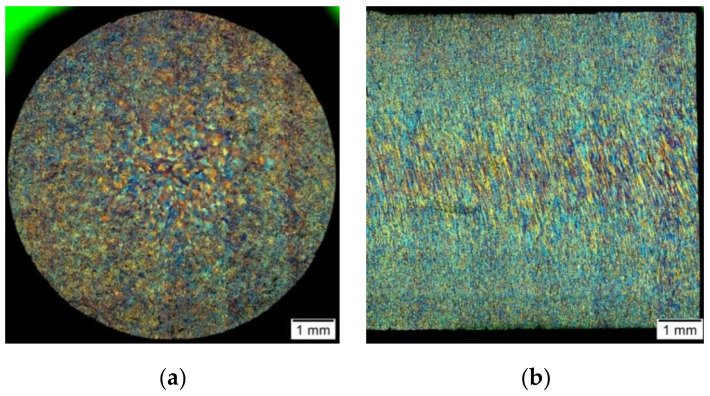
Microstructure of the 5019 aluminium alloy after plastic deformation: (**a**) cross-section; (**b**) longitudinal section.

**Figure 20 materials-14-03508-f020:**
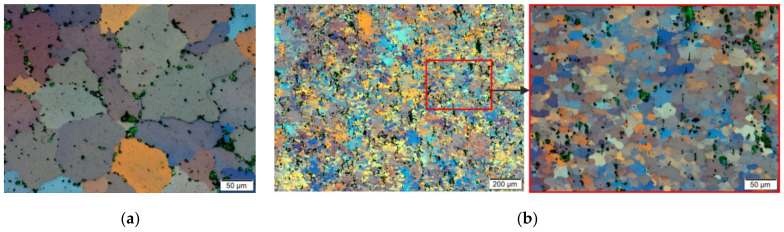
Microstructure of the 5019 aluminium alloy after plastic deformation: (**a**) centre of the sample; (**b**) edge of the sample; cross-section, magnification 200×.

**Figure 21 materials-14-03508-f021:**
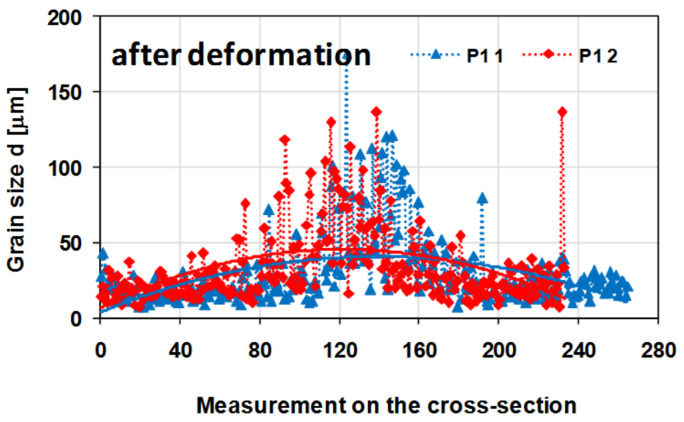
Changes in the grain size on the cross-section of a sample from the 5019 aluminium alloy after the deformation process; the measurements were made in two perpendicular directions (as shown in Figure 15); P1—material condition after hot torsion.

**Figure 22 materials-14-03508-f022:**
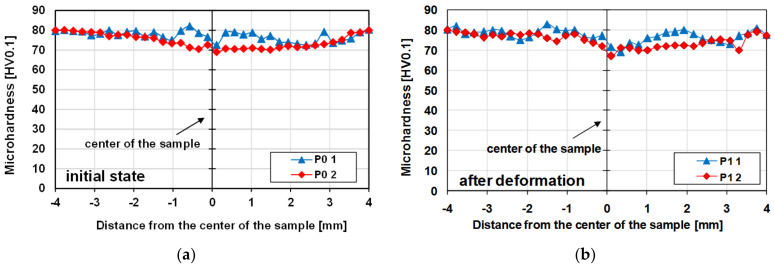
Changes in microhardness on the cross-section of the 5019 aluminium alloy samples (measurements made in two perpendicular directions—according to Figure 14): (**a**) undeformed material; (**b**) material after plastic deformation.

**Figure 23 materials-14-03508-f023:**
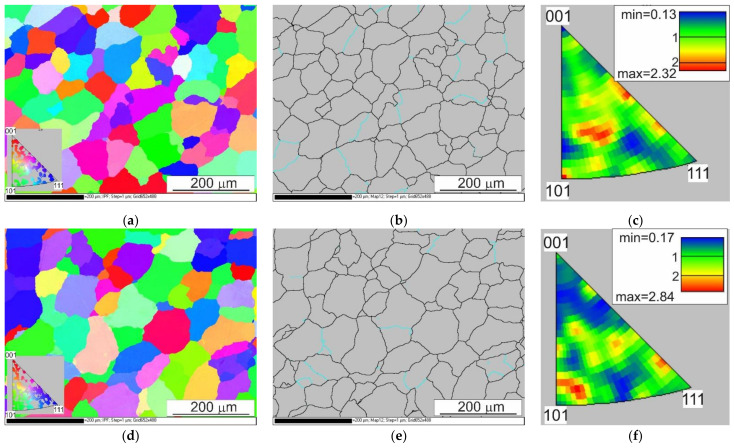
EBSD analysis results of the sample after homogenisation; (**a,b,c**) centre of the sample, (**d,e,f**) distance from the centre = 0.67 r, (**g,h,i**) distance of the centre = 0.72 r; (**a,d,g**) EBSD maps showing changes in orientation, (**b,e,h**) maps showing the types of edges (the edges of a large angle are marked in black, and the edges of a small angle are marked in blue) and a (**c,f,i**) basic triangle—orientation intensity.

**Figure 24 materials-14-03508-f024:**
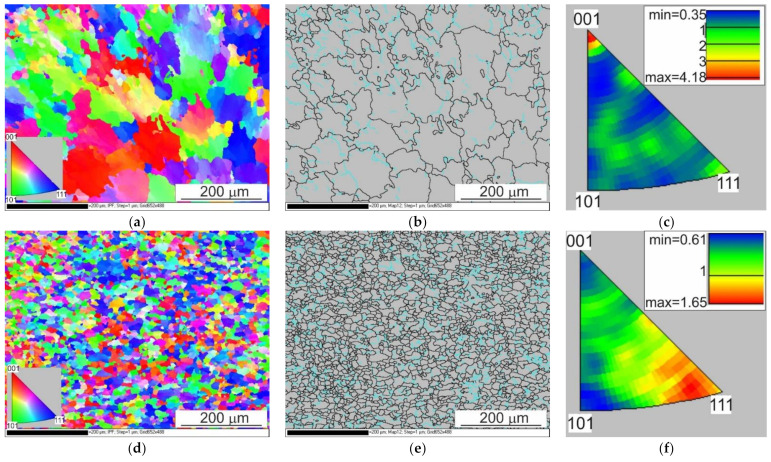
EBSD analysis results of the sample after deformation: (**a,b,c**) centre of the sample, (**d,e,f**) distance from the centre = 0.67 r, (**g,h,i**) distance of the centre = 0.72 r; (**a,d,g**) EBSD maps showing changes in orientation, (**b,e,h**) maps showing the types of edges (the edges of a large angle are marked in black, and the edges of a small angle are marked in blue) and a (**c,f,i**) basic triangle—orientation intensity.

**Figure 25 materials-14-03508-f025:**
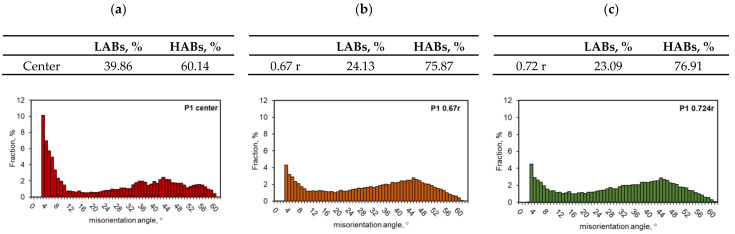
Changes in disorientation angles after hot torsion: (**a**) sample centre, (**b**) at a distance from the centre of 0.67 r and (**c**) at a distance from the centre of 0.72 r.

**Figure 26 materials-14-03508-f026:**
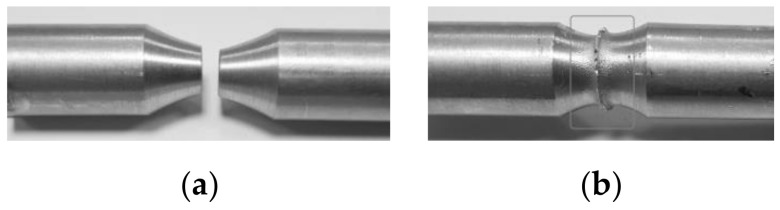
6XXX series aluminium alloy sample: (**a**) before the friction welding process; (**b**) after the friction welding process.

**Figure 27 materials-14-03508-f027:**
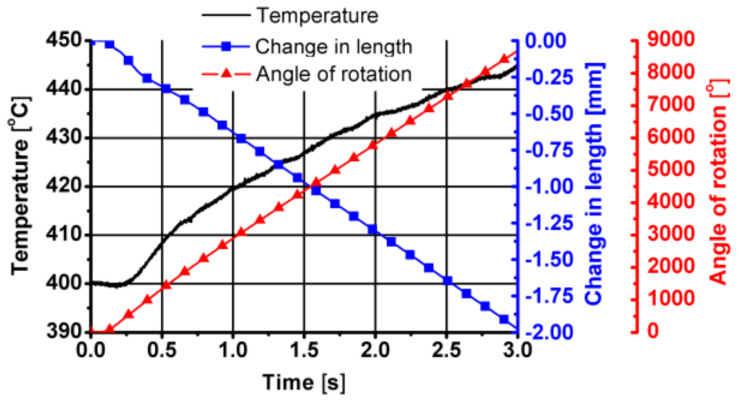
Temperature, length and angle of rotation changes over time in the first stage of friction welding of the aluminium 6XXX series—simultaneous torsion with compression.

**Figure 28 materials-14-03508-f028:**
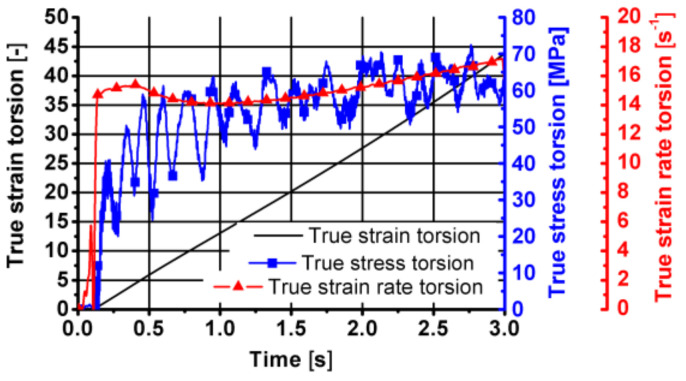
Changes in the true strain torsion, true stress torsion and true strain rate torsion over time in the first stage of friction welding of the 6XXX series aluminium—simultaneous torsion with compression components resulting from torsion.

**Figure 29 materials-14-03508-f029:**
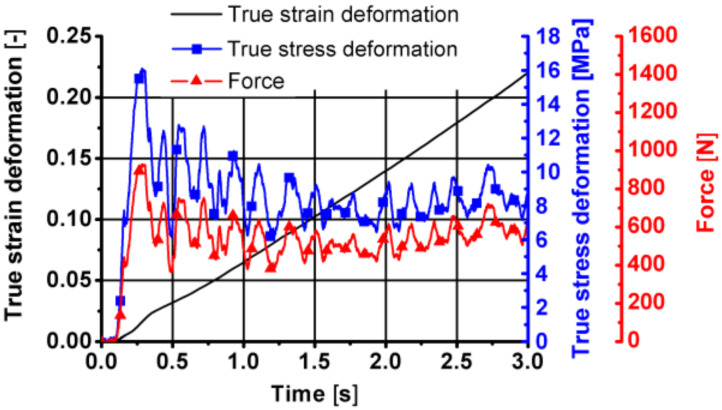
Changes in the true strain deformation, true stress deformation and force over time in the first stage of friction welding of the aluminium 6XXX series—simultaneous torsion with compression components resulting from compression.

**Figure 30 materials-14-03508-f030:**
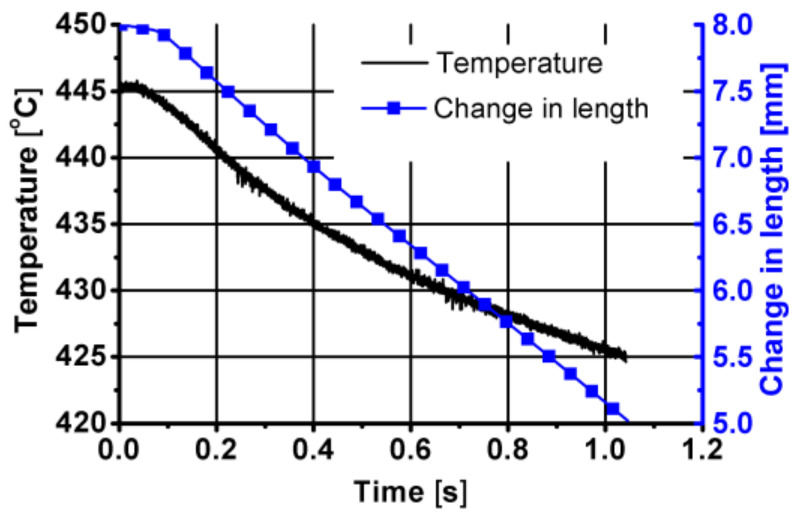
Temperature and length changes over time in the second stage of the friction welding of the 6XXX series aluminium—compression.

**Figure 31 materials-14-03508-f031:**
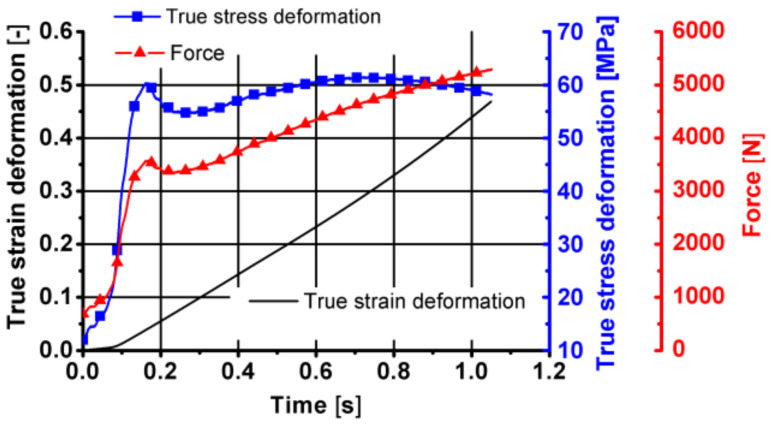
Changes in the true strain deformation, true stress deformation and force over time in the second stage of the friction welding of the 6XXX series aluminium—compression.

**Table 1 materials-14-03508-t001:** Coefficients of Equation (4) used to determine σ_p_ values of the 5019 aluminium alloy (Variant 1) [45].

A	m_1_	m_2_	m_3_	m_4_	m_5_	m_7_	m_8_	m_9_
0.271553	−0.00957753	−0.0823773	**−0.246413**	**−0.000573716**	**0.000622463**	**−0.164103**	0.00101501	1.65143

**Table 2 materials-14-03508-t002:** Coefficients of Equation (4) used to determine σp values of the 5019 aluminium alloy (Variant 2) [45,46].

A	m1	m2	m3	m4	m5	m7	m8	m9
0.271553	−0.00957753	−0.0823773	**−0.2465**	**−0.002**	**0.0001**	**−0.032**	0.00101501	1.65143

## Data Availability

The data presented in this study are available on request from the corresponding author.

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
