# Peer review of "Theoretical and Experimental Analysis of the Hot Torsion Process of the Hardly Deformable 5XXX Series Aluminium Alloy"

_materials, 2021, doi:10.3390/ma14133508_

Round 1

Reviewer 1 Report

The paper is an important  contribution in the field of numerical and physical modelling of hot torsion of a hardly deformable 5XXX series aluminium alloy.  A consistent analysis of the current stage is made and the principal achievements from the literature are evidentiate in chapter 1. In chapter2, authors present the numerical and experimental methods used in their analysis. For numerical analysis they are using the finite element software FORGE. The theoretical aspects of the program are presented in connection with the hot torsion modelling. For experimental part, the STD 812 torsion plastometer is used and the main parameters of the device are presented. For metallographic analysis and microhardness measurements, top equipments are used.

Starting with the page 7 till the page 20, a deeply analysis of both numerical and experimental results is made. The analysis shows the impact of the accuracy of the mathematical model of rheological properties toward the distribution and values of strain and stress intensity of the tested aluminium alloy. And this model is in correlation with the proper determination of the strain parameters distribution and the stress intensity in the experimental torsion test.

On the other hand, an improper microstructure on the cross-section after the hot torsion is proved. The level of microhardness of the material after the hot torsion process is discussed. The conclusions are  clear  and very well formulated.

Reviewer 2 Report

Recommendations to authors:

*Use established symbols to image description (see line 273, 307, 326, 329 etc. change Rys./Fig etc.).

Reviewer 3 Report

The manuscript is focused on the theoretical and experimental analysis of hot torsion test of an aluminium alloy. The influence of the accuracy of the mathematical model of rheological properties on the strain parameters and the yield stress was determined. The influence of the strain parameters on the grain size distribution and the microhardness was also focused.

The paper is very interesting and fits the aim and scope of the journal.

Problems:

Materials and Methods

The material of the test specimens is the 5019 aluminium alloy? The material should be identified.

How did you ensure homogeneous temperature on the test specimen? The induction heating process is characterized to be localized.

The hot torsion test follows the standard? The technical drawing of the test specimen should be added to the manuscript.

Results

Can you explain the determination of the coefficients of equation (4) (table 3 and 4) and the differences between the table 3 and 4?

For obtain the temperature distribution of the figure 5, the induction heating process was numerically simulated?

The perspective view of figure 6(c) should be added on figure 5 to improve the interpretation of the temperature distribution.

Round 2

Reviewer 3 Report

Accepted.